# High-fidelity qutrit entangling gates for superconducting circuits

**Noah Goss** [1,2] ✉, **Alexis Morvan**[2], **Brian Marinelli**[1,2], **Bradley K. Mitchell**[1], **Long B. Nguyen** [2], **Ravi K. Naik**[1,2], **Larry Chen** [1], **Christian Jünger** [2], **John Mark Kreikebaum**[1,3], **David I. Santiago**[2], **Joel J. Wallman** [4] & **Irfan Siddiqi**[1,2,3]

Ternary quantum information processing in superconducting devices poses a promising alternative to its more popular binary counterpart through larger, more connected computational spaces and proposed advantages in quantum simulation and error correction. Although generally operated as qubits, transmons have readily addressable higher levels, making them natural candidates for operation as quantum three-level systems (qutrits). Recent works in transmon devices have realized high fidelity single qutrit operation. Nonetheless, effectively engineering a high-fidelity two-qutrit entanglement remains a central challenge for realizing qutrit processing in a transmon device. In this work, we apply the differential AC Stark shift to implement a flexible, microwave-activated, and dynamic cross-Kerr entanglement between two fixed-frequency transmon qutrits, expanding on work performed for the *ZZ* interaction with transmon qubits. We then use this interaction to engineer efficient, high-fidelity qutrit CZ† and CZ gates, with estimated process fidelities of 97.3(1)% and 95.2(3)% respectively, a significant step forward for operating qutrits on a multi-transmon device.

Quantum error correction (QEC)[1] is necessary for noisy intermediate-scale quantum (NISQ)[2] computers to realize their full potential. The surface code[3,4] using qubits is considered the main route to fault tolerance[5–8], though its technical challenges have led the community to explore other approaches that could have more favorable QEC schemes, such as storing a two level system in the large Hilbert space of quantum oscillators[9,10]. Another alternative is to use *d*-dimensional quantum objects, or qudits, which mobilize a larger and more connected computational space than their qubit counterparts. Qutrits, the simplest form of qudits, can provide advantages in QEC for magic state distillation[11,12], compactly encoding qubits[13], and can be used to encode logical qutrits themselves[14–16]. Additionally, there are several proposals utilizing qutrits to improve quantum applications such as factoring with Shor's algorithm[17], performing the quantum Fourier transformation[18], providing speedups for oracle based quantum algorithms[19], improving quantum simulation[20], and asymptotically

improving algorithms such as Grover's search[21,22]. Realizing multi-qudit systems, however, is challenging due to the complexities of the larger Hilbert space. Nonetheless, coherent control of qudits has been performed in several physical platforms[23–28]. While state of the art experiments have demonstrated high-fidelity qudit entangling gates with trapped ions[23,24] and photonic circuits[26], generating high-fidelity, maximally entangling two-qudit gates remains a major challenge in superconducting circuits.

The most commonly used qubit in superconducting circuits[29], the transmon[30], is well suited to be operated as a qutrit due to its weak anharmonicity. Technical advancements in microwave engineering and improved fabrication techniques have increased transmon coherence times[31], enabling coherent control of the full qutrit Hilbert space. Furthermore, dispersive readout can be used for high-fidelity single shot qutrit readout[27]. In addition, high-fidelity single qutrit operations[32,33], quantum information scrambling[28], compact

[1]Department of Physics, University of California, Berkeley, Berkeley, CA 94720, USA. [2]Computational Research Division, Lawrence Berkeley National Laboratory, Berkeley, CA 94720, USA. [3]Materials Science Division, Lawrence Berkeley National Laboratory, Berkeley, CA 94720, USA. [4]Keysight Technologies Canada, Kanata, ON K2K 2W5, Canada. ✉e-mail: noahgoss@berkeley.edu

decompositions of multi-qubit gates[34–37], and improved qubit readout[38] have all been demonstrated using transmons as qutrits. Nonetheless, past qutrit entangling gates in transmons have been limited by relying on either a slow, static interaction which can only be sped up at the expense of increased quantum crosstalk on the qutrit processor or an interaction that restricts the entanglement to only a subspace of the qutrit.

In this work, we characterize the differential AC Stark shift[39–42] on two fixed frequency transmon qutrits with static coupling and leverage it to generate dynamic qutrit entangling phases. The tunable nature of this entangling interaction enables a large on/off ratio, allowing for future high-fidelity, simultaneous single-qutrit and two-qutrit operations in transmon qutrit processors. With this interaction, we engineer the ternary controlled-Z gate (CZ) and its inverse (CZ†). Both gates performed in our work are universal for ternary computation, maximally entangling, and Clifford gates needed for QEC in qutrits. We achieve an estimated process fidelity of 97.3(1)% and 95.2(3)% for the CZ† and CZ respectively, measured using cycle benchmarking[43] and our generalization of the cross-entropy benchmarking routine[44]. The fidelity of the CZ† represents a factor of 4 reduction in infidelity over previous two qutrit transmon gates[28,32]. Finally, we numerically demonstrate that our gate scheme is efficient for generating additional two-qutrit Clifford gates.

## Results

### Differential AC stark shift

Recent works by refs. [39–42] demonstrated that the architecture employed in the Cross-resonance (CR) entangling gate can also realize a two-qubit CZ gate by leveraging the conditional Stark shifts from simultaneously driving a pair of coupled qubits off-resonantly. The advantages of this method are two-fold: firstly, the CZ gate commutes with $ZZ$ errors from the always-on dispersive coupling between the transmons. Secondly, unlike in the CR gate, the frequency of the microwave drive can take a range of values. This flexibility in drive frequency affords significant advantages in avoiding frequency collisions with other transmons or spurious two-levels systems[45]. The generalization of conditional Stark shifts to qutrits is straightforward. Working in the energy eigenbasis of our two qutrit Hilbert space, up to single-qutrit phases, one's system evolves under the cross-Kerr Hamiltonian:

$$\mathcal{H} = \alpha_{11}|11\rangle\langle 11| + \alpha_{12}|12\rangle\langle 12| \\ + \alpha_{21}|21\rangle\langle 21| + \alpha_{22}|22\rangle\langle 22|, \tag{1}$$

where each term can be be calculated with perturbation theory (see Supplementary Note 2). In this microwave-activated case, the $\alpha_{ij}$ ($ZZ$-like) terms are given by:

$$\alpha_{ij} = A_{ij}(\omega_d)\Omega_a\Omega_b\cos(\phi_a - \phi_b), \tag{2}$$

where $\Omega_i$ and $\phi_i$ are respectively the amplitude and phsae of the drive on transmon $i$. The coefficients $A_{ij}$ are functions of the proximity of the microwave drive frequency ($\omega_d$) to nearby transitions. We note that this Hamiltonian generates entanglement between the entire two-qutrit Hilbert space, contrary to the CR case, where the entanglement is mostly restrained to a subspace of the qutrit[28,46]. This dynamic, driven cross-Kerr interaction is depicted schematically in Fig. 1. It is important to note that the number of degrees of freedom in this interaction are not sufficient, in general, to realize a Clifford two-qutrit gate like the CZ gate with a single round of cross-Kerr entanglement, a difficulty discussed in further detail in the next section.

Measuring the $ZZ$ interaction in the qubit case can be performed by a conditional Ramsey experiment or through a dynamically decoupled JAZZ (Joint-Amplification-of-$ZZ$) sequence that removes the low frequency drift[47,48]. In the larger Hilbert space of two qutrits, we

need to measure four of these entangling phases with a rate of accumulation set by $\alpha_{ij}$ in Eq. (2). To simplify the measurement and reduce the number of experiments needed, we generalize the controlled-Ramsey experiment to the full qutrit space with a pulse sequence presented in Fig. 2a. In this sequence, we apply simultaneous ternary Hadamard gates on both qutrits, execute the microwave drive, and subsequently perform the full two qutrit state tomography. Doing so for several durations of the Stark driving allow us to fully characterize the conditional and unconditional Stark shifts.

We demonstrate in Fig. 2b the result of this measurement scheme: the entangling phases increase linearly with the duration of the Stark drive, where the proportionality constant is set by $\alpha_{ij}$ as predicted by our cross-Kerr model in Eq. (1). In Fig. 2c, d, we present how the driven cross-Kerr interaction depends on the parameters of our entanglement scheme, specifically the phase and the amplitude of the Stark drive. We note that the qualitative behavior is properly captured by the perturbation theory in Eq. (2). We also explore the behavior as a function of the drive frequency in Fig. 2e. In this case, the perturbation theory fails, but an ab-initio master equation simulation captures some of the response; additional details on the frequency dependence of all $\alpha_{ij}$ terms and the master equation simulation can be found in the supplement. We expect the unaccounted features can be attributed to higher-order terms, frequency dependent classical crosstalk, or parasitic two level systems (TLS) in our device. In an experimental setting, the flexibility of this entanglement allows us the freedom to set the drive frequency far from any of these features.

### Qutrit CZ/CZ† gate

We next construct qutrit controlled-phase gates utilizing this entangling interaction. The qutrit CZ and CZ† gate are both maximally entangling and members of the two-qutrit Clifford group making them particularly useful gates for ternary computation. The CZ gate is defined as:

$$U_{CZ} = \sum_{i,j\in\{0,1,2\}^2} \omega^{ij}|ij\rangle\langle ij|, \tag{3}$$

with $\omega = e^{2i\pi/3}$, the third root of unity; the CZ unitary follows directly from generalizing the qubit Pauli group to qutrits and is explained in

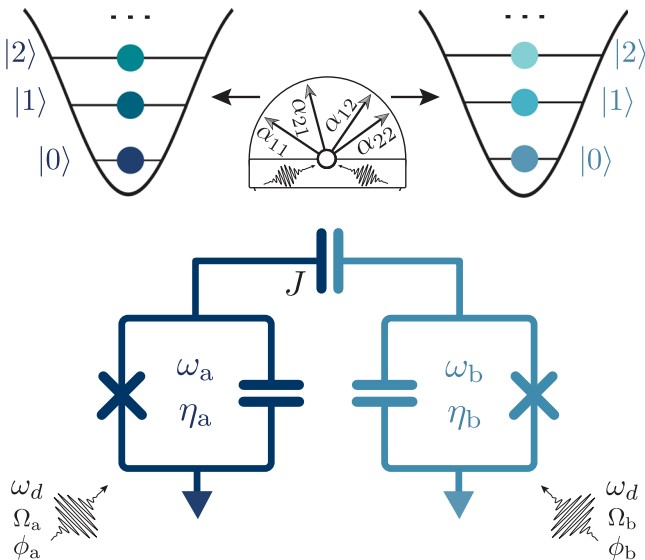

**Fig. 1 | Microwave-activated cross-Kerr entanglement.** Two transmon qutrits with qubit frequency $\omega_i$, anharmonicity $\eta_i$, and coupling $J$, experience a dynamical cross-Kerr ($ZZ$-like) entanglement when simultaneously driven by an off-resonant microwave drive. The strength of the cross-Kerr entangling terms ($\alpha_{11}, \alpha_{12}, \alpha_{21}, \alpha_{22}$) is tuned by the parameters of the microwave drive ($\omega_d, \Omega, \phi$).

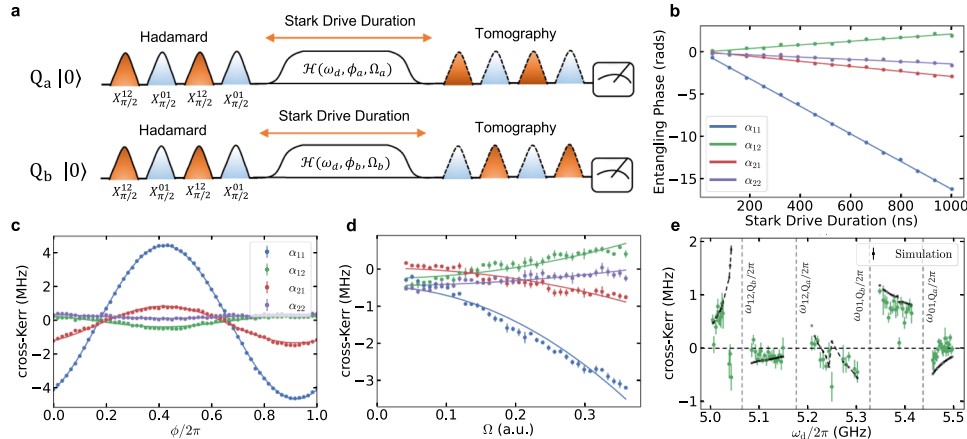

**Fig. 2 | Characterizing the dynamical cross-Kerr entanglement. a** To study the accumulation of entangling phases under the driven cross-Kerr interaction, we place two qutrits in a full superposition using ternary Hadamard gates (virtual Z gates ommited in diagram), then study the evolution under the Stark drive scheme by performing state tomography. **b** We demonstrate fitting the accumulation of entangling phase found by tomography to our linear, driven cross-Kerr model, where $\alpha_{ij}$ is the slope of the line and the uncertainty is from the linear fit. **c**, **d**, We match the behavior of the cross-Kerr entanglement given relevant experimental parameters in our system to our Hamiltonian model for the relative phase of the driving, $\phi$, and amplitude of the driving, fixing $\Omega = \Omega_a = \Omega_b$. **e** We additionally compare the dependence of $\alpha_{12}$ on the frequency of the drive $\omega_d$ using an ab-initio master equation simulation in QuTiP[56,57].

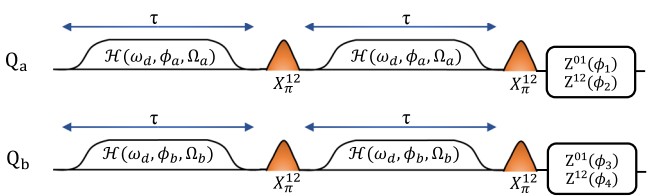

**Fig. 3 | Gate schematic.** For the CZ and CZ$^\dagger$ gate, we perform two rounds of cross-Kerr entanglement for duration $\tau$ with interleaved echo pulses in the $\{|1\rangle, |2\rangle\}$ sub-space which shuffle the entangling phases. For proper conditions on the $\alpha_{ij}$ terms in Eq. (1), the CZ$^\dagger$(CZ) is compiled with a total gate time of 580(783) ns. The local Z terms in both two level subspaces of the qutrit are then undone using virtual Z gates.

further detail in Supplementary Note 3. Under simultaneous Stark drives, the two-qutrit Hilbert space follows the unitary evolution $U = \exp(-i(\mathcal{H} + \phi_1 I \otimes Z^{01} + \phi_2 I \otimes Z^{12} + \phi_3 Z^{01} \otimes I + \phi_4 Z^{12} \otimes I)\tau)$ where $\mathcal{H}$ is given in Eq. (1). To perform a CZ gate with a single round of cross-Kerr driving, for example, one would need to find driving parameters meeting the conditions: $\{\alpha_{11} = \alpha_{22} = 2\alpha_{21} = 2\alpha_{12}\}$, a task that is not broadly feasible. Practically speaking, we desire a compromise between the most general and robust gate scheme and this "fine-tuned" approach, while still taking advantage of the dynamical nature of our cross-Kerr interaction. By employing the pulse scheme in Fig. 3, where echo pulses in the $\{|1\rangle, |2\rangle\}$ subspace shuffle entangling phases, we have the modified unitary evolution (omitting the single-qutrit phases for brevity):

$$U = \exp(-i[(\alpha_{11} + \alpha_{22})\tau(|11\rangle\langle 11| + |22\rangle\langle 22|) \\ + (\alpha_{12} + \alpha_{21})\tau(|12\rangle\langle 12| + |21\rangle\langle 21|)]), \quad (4)$$

Using the experimental knobs demonstrated in Fig. 3, we are able to satisfy approximate conditions on our cross-Kerr evolution that are ideal for compactly generating qutrit controlled phase gates. Specifically, we find drive parameters that satisfy $(\alpha_{11} + \alpha_{22}) \approx -(\alpha_{12} + \alpha_{21})$ for performing the CZ gate and $(\alpha_{11} + \alpha_{22}) \approx 2(\alpha_{12} + \alpha_{21})$ for performing the CZ$^\dagger$ gate. The Stark drive parameters that meet these conditions are provided in Supplementary Note 1. Under these conditions on the cross-Kerr, and with the unitary evolution provided by our gate scheme in Eq. (4), at some drive time $\tau$, we will have approximately acquired the desired entangling phases found in Eq. (3) to synthesize

respectively a CZ or CZ$^\dagger$ unitary. To ensure adiabaticity and limit leakage, we perform the Stark drive via flat-top cosine pulses with ramp up and down features. This ramp leads to effective offsets on the accumulation of the entangling phases, as we only expect the linear accumulation of entangling phase given by $\alpha_{ij}$ in Eq. (2) to correspond to the flat-top of the Stark drive. When tuning up the two-qutrit gates, this means that we first perform parameters sweeps to find regions where the previously mentioned conditions on the $\alpha_{ij}$s are met, then perform the actual pulse scheme in Fig. 3 and adjust the Stark drive parameters until the target entangling phases given by Eq. (3) are most accurately acquired. Finally, as outlined schematically in Fig. 3, one can undo the local Z phases (found via tomography) in both the $\{|0\rangle, |1\rangle\}$ and $\{|1\rangle, |2\rangle\}$ subspaces with virtual Z gates[49]. In this work, we performed the CZ and CZ$^\dagger$ on two different pairs of transmon qutrits, demonstrating the flexible nature of generating two-qutrit gates from this driven cross-Kerr scheme.

## Benchmarking

We first benchmark our two-qutrit gates with cycle benchmarking (CB)[43] using True-Q[50]. While originally written in terms of qubits, CB naturally generalizes to qutrits[32]. We use CB instead of, e.g., interleaved randomized benchmarking[51,52], because it requires significantly fewer multi-qutrit gates per circuit. We describe the generalization in Supplementary Note 6. With this technique, we estimate the Weyl (generalized-Pauli) error rate of the CZ$^\dagger$ and CZ gate to be 2.7(1)% and 4.8(3)% respectively. By contrast, the highest fidelity, two-qutrit gate performed previously with transmons had an error rate of 11.1%[28]. CB also allows us to construct the Weyl-twirled error per channel[32] of the unitary in Fig. 4. This provides us with an estimate of the worst case scenario of less than 8% and demonstrates a relatively low dispersion of our error channels.

As an added confirmation of the fidelity of the CZ$^\dagger$ gate, we generalize the cross-entropy benchmarking (XEB) routine[44,53] to tailor all gate errors into a depolarizing channel, for work with qutrit unitaries. In the qutrit case, we find sufficient tailoring of our noise can be performed by interleaving random SU(3) gates around our target gate. The circuit diagram for qutrit XEB is in Fig. 4b and the results can be found in Fig. 4e. We find that the depolarized fidelity of the CZ$^\dagger$ dressed with random SU(3) gates agrees with the estimate of the process fidelity from our Weyl twirled CB results within a standard error. Additional discussion of the qutrit XEB method is provided in Supplementary Note 5.

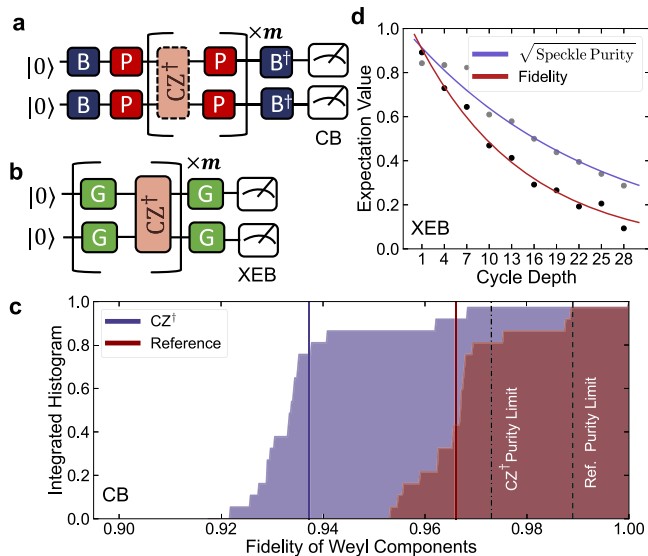

**Fig. 4 | Benchmarking. a** Circuit schematic of cycle benchmarking (CB). The errors of the CZ† are twirled via random Weyl gates (red) to tailor errors into stochastic Weyl channels. The initial state and measurement basis (blue) are selected to pick out the decay associated with specific Weyl channels. **b** Circuit schematic of cross-entropy benchmarking (XEB). The errors of the CZ† are twirled via random SU(3) gates (green) to tailor the noise to a simple depolarizing channel. **c** An integrated histogram of CB for both the CZ† gate and a reference cycle, with the solid vertical lines giving the fidelities 0.936(1) and 0.966(1) respectively, yielding an estimated process fidelity of 97.3(1)%. We extract an error budget directly from CB, estimating a purity limited fidelity of 0.973(9) and 0.989 (with negligible error) for the dressed CZ† and reference cycles, yielding a purity limit 0.986(9) for the isolated CZ† gate. **d** From XEB we estimate the depolarized fidelity as 0.933(3). Additionally, we estimate the speckle-purity limited fidelity of the CZ† dressed with random SU(3) gates to be 0.961(3).

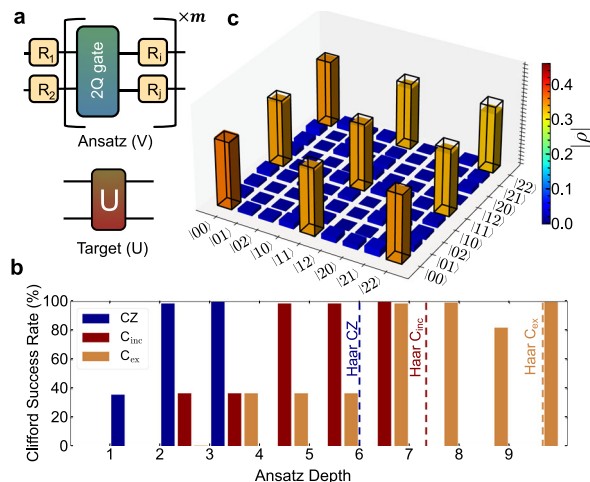

**Fig. 5 | Demonstration of gate expressability. a** A parameterized Ansatz circuit (V) is used to synthesize a target unitary (U), given some 2-qutrit gate and arbitrary SU(3) gates. **b** We study the Ansatz circuit (V) for the different two-qutrit gates discussed in the text for 1000 Haar random and Clifford gates, minimizing the infidelity $1 - \mathcal{F}(V,U)$. The dashed lines represent 100% numerical success for synthesizing our set of Haar random gates, and the bars display the success rate for synthesizing Clifford gates. We perform the minimization until we find a 100% success rate for each two-qutrit gate between depths $0 \leq m \leq 9$. **c** An experimentally reconstructed density matrix of the two qutrit Bell state $|\psi\rangle = \frac{1}{\sqrt{3}}(|00\rangle + |11\rangle + |22\rangle)$ formed using a single CZ gate with state fidelity $\mathcal{F} = 0.952$. The black outline is the target density matrix.

Finally, we would like to be able to characterize what fraction of the errors present in our two qutrit unitaries are coherent on the time scale of multiple experiments, and thus could be removed by improved calibration. As we show in supplementary note 6, under the depolarizing unitary noise model, the variance of CB and XEB circuits both provide a robust method of estimating the purity limit[54,55]. The corresponding estimates are shown in Fig. 4c with the CB estimate of 97.3(9)% exceeding the speckle-purity limit of 96.1(3)% for the dressed CZ† gate. This disagreement is likely due to the fact that the CB data reveals that the noise is dominated by single-qutrit phase errors. As these errors are likely to fluctuate around a mean, they will add dephasing errors that are not captured by the depolarizing unitary model used to analyze XEB. Another possible contributing factor is that the noise fluctuated between the XEB and CB experiments, which were performed in separate batches.

**Gate synthesis of two qutrit unitaries**

To study the expressibility of the two-qutrit gates in this work, we numerically explore the ability of the ternary CZ/CZ† (localy equivalent to each other and the CSUM gate) to synthesize other two-qutrit gates, and compare them to two-qutrit entangling gates that only entangle a subspace of the qutrit, such as the controlled-exchange ($C_{ex}$) and controlled-increment ($C_{inc}$) gates performed on a trapped ion system in ref. 24. To this end, we consider an Ansatz circuit V as in Fig. 5a, with depth m, which we use to synthesize target circuits belonging to either the two qutrit Clifford group or set of Haar random gates. The gate synthesis is performed by optimizing the ansatz parameters to minimize the distance between V and U, i.e. the infidelity $1-\mathcal{F}(V, U)$.

We perform this numerical investigation on 1000 Haar random gates and 1000 Clifford gates. We find that all 1000 Haar random gates

can be synthesized at depth 6 for CZ/CZ†, 7 for $C_{inc}$, and 9 for $C_{ex}$. The synthesis success rate for all 3 unitaries in terms of target Clifford circuits are shown in Fig. 5b. Notably, almost all two qutrit Clifford gates were successfully compiled at depth 2 in CZ/CZ†, with 100% success at depth 3. By contrast, $C_{ex}$ and $C_{inc}$ did not demonstrate as much improvement at synthesizing Clifford gates over Haar random gates, achieving 100% success for target Clifford gates at depth 6 for $C_{inc}$ and 9 for $C_{ex}$. Additionally, unique amongst these gates, the CZ/CZ† can generate maximally entangled two qutrit states with a single iteration of the gate. We demonstrate the power of this feature in the experimentally reconstructed qutrit Bell-state density matrix in Fig. 5c. In summary, the maximally entangling CZ/CZ† gates have low intrinsic errors and can also can synthesize a very important family of gates for QEC (the Cliffords)[1] at much lower depths than the two-qutrit gates which only entangle a subspace of the qutrit.

## Discussion

We realized a microwave-activated, dynamic cross-Kerr entangling interaction that can be employed to engineer qutrit entangling phases with high precision. Leveraging this interaction, we generated two maximally entangling and high-fidelity two-qutrit gates on two separate pairs of fixed-frequency transmon qutrits. We demonstrated numerically that these two qutrit gates are efficient for producing additional two-qutrit unitaries, especially other Clifford gates. Future work developing a systematic gate tune up procedure may prove essential in improving the fidelity and scalability of our approach. Additionally, a study of the effects of this gate scheme on spectator qutrits will also be necessary for determining its scalability. We expect that by being maximally entangling and a member of the two-qutrit Clifford group, the gates performed in this work will prove especially powerful in future efforts to employ qutrits for QEC, quantum simulation, and quantum computation. Perhaps most importantly, all of this work was performed on multi-transmon quantum processors which are normally used for qubit experiments; the untapped potential of transmons as qutrits is only beginning to be explored. As a final

note, the two-qutrit Hilbert space is larger than even the three qubit Hilbert space; as the community continues to explore qudits, we propose that metrics and benchmarks should be developed to reasonably compare qudit vs. qubit gates.

## Data availability

Source data are provided with this paper. All other data that supports the findings of this study are available from the corresponding author upon reasonable request.

## Code availability

Cycle Benchmarking and the expressability analysis were performed using properietary TrueQ™ software (https://trueq.quantumbenchmark. com). All other code that supports the findings of this study is available from the corresponding authors on reasonable request.

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

## Acknowledgements

We are grateful to A. Hashim, W. Livingston, and I. Hincks for conversations and insights. This work was supported by the Quantum Testbed Program of the Advanced Scientific Computing Research Division, Office of Science of the U.S. Department of Energy under Contract No. DE-AC02-05CH11231.

## Author contributions

N.G. and A.M. conceived and planned the experiment. N.G., A.M., B.M. and B.K.M. performed the experiment. N.G., B.M. and J.J.W. analysed the data. L.B.N. and R.K.N. contributed to the analysis and discussion of the results. L.C., C.J. and J.M.K. fabricated the transmon devices. N.G. wrote the manuscript with assistance from A.M., J.J.W., B.M., L.B.N., and R.K.N. All work was carried out under the supervision of D.I.S. and I.S.

## Competing interests

J.J.W. has a financial interest in Keysight technologies and the TrueQ™ software. The remaining authors declare no competing interests.
