## [Peer Review File · Nature Communications]

REVIEWER COMMENTS

Reviewer #1 (Remarks to the Author):

This manuscript reports on high-fidelity qutrit entangling gates for superconducting transmon circuits. A microwave-activated cross-Kerr interaction is used to realize the qutrit entangling gates. The reported results constitute a good demonstration of logical operations on superconducting qutrits.

My biggest reservation about the manuscript is that the main achievement is quantitative rather than qualitative. Some of the authors of this manuscript have previously reported qutrit entangling gates in references 28 and 32. In [28], the lowest infidelity was 11.1%. In this manuscript the lowest infidelity is 2.7%. This is an improvement. However, it is a quantitative improvement. It is true that the gate mechanism is different. However, this is not a major advance in the field.

Another concern is that qutrit logic is a niche research area. Most research in the field is focused on qubits because of obvious advantages such as simplicity and better coherence.

Overall the manuscript is clearly written. However, there are some poorly written parts. Here are some examples. It is not explained what “more connected” means. The work is motivated by saying that realizing multi-qudit systems is challenging. However, if transmons are qudits, then any multi-transmon circuit is a multi-qudit system. So this motivation sounds weak. The manuscript says, without a proper justification, that the interaction in this work “generates entanglement between the entire two-qutrit Hilbert space”. The poor wording aside, there is no explanation for how this is different from previous work on two-qutrit gates. These problems can be fixed with a rewriting of the poorly written parts.

A technical point about the qutrit CZ gate is that it does not reduce to a qubit CZ gate if we look only at the states $0/1$. So the basis for generalizing the CZ gate from qubits to qutrits warrants some justification. When equation 3 is introduced, it is not clear if it is taken from the literature or introduced for the first time in this manuscript.

Except for these issues, the manuscript reports good results in the area of superconducting qutrits.

Reviewer #2 (Remarks to the Author):

In the manuscript by Goss et. al., the authors demonstrated and benchmarked an all-microwave qutrit CZ gate using differential AC Stark shifts. This paper has many impressive aspects including a detailed experimental characterization of the cross-Kerr dynamics, and a clever use of echoes to construct the qutrit CZ gate without fine tuning the cross-Kerr dynamics. This work is an extension of differential AC Stark shifts in qubit systems [ref. 39-42]. The ability to generate 3-dimensional Bell states using a single qutrit CZ gate is attractive; potentially allowing one to generate 3-dimensional multipartite entangled GHZ states in transmons. This work deserves publication, but I'd like to see the following questions addressed first.

1. Can the authors comment on how the charge noise in the 2 state impact the qutrit CZ gate? In the device data given in the supplementary, the T2 times for 12 and 02 levels are considerably lower than that of 01 levels.

2. I did not find a centralized description of the gate parameters for the qutrit CZ/CZ[†] gate. The gate times are in the caption of Fig 3, and the powers in the supplementary. What was the drive frequency and how were the frequency and power chosen?

3. Did the authors simulate terms beyond the 4 in the cross-Kerr Hamiltonian? In regions where the differential AC Stark fails the cross-Kerr Hamiltonian is presumably no longer a good approximation and other terms will be present.

4. Beyond the 4 resonances present in Fig. 2e, are there additional regions where the qutrit CZ gate does not perform well? For instance near the middle between $\omega_{01,a}$ and $\omega_{01,b}$ I would expect CZ dynamics to get complicated due to 2-photon transitions.

5. In the paragraph below equation 4, the 2 approximated equations seem to imply $\alpha_{11} + \alpha_{22} \sim \alpha_{12} + \alpha_{21} \sim 0$. Can the authors double-check this?

6. In the same paragraph the authors state “the ramp feature can lead to offsets ... approximate relationship on the α_{ij} terms with some adjustment necessary”, what is the adjustment?

7. It might be helpful to have different labels for the phases ϕ_a ϕ_b and powers in Fig.2a and 3, to be consistent with Eq. 2 and Fig. 1.

8. Why were the CZ and CZ[†] gates demonstrated on different samples?

Reviewer #3 (Remarks to the Author):

Review of High-Fidelity Qutrit Entangling Gates for Superconducting Circuits

The authors implement High fidelity qutrit entangling gate that can provide the basis for qutrit-based quantum computers. This gate is implemented using a cross-Kerr resonant pulse. The gate designed appears to have higher entangling power than a Cinc or Cex gate. The work does establish significance for the field. It provides a proof of principle demonstration of a powerful entangling gate for qutrits. This demonstration of the gate likely can be used to help develop more efficient compilers for qubit architecture simulations such as those provided in Gokhale et al. "Asymptotic improvements for quantum circuits via qutrits." This work appears to be better than 82% fidelity result for a Cinc gate implemented by Morvan et al. in 2020.

I find the authors work novel and timely and warrants publication following the minor revisions.

Requested Revisions:

I have a few revisions that I believe are necessary before approving for publication

1.) I would like to see a slightly wider discussion of the benefits qutrit architecture may apply to quantum algorithms. As the current connection to the benefits of qutrit hardware to algorithms seem sparse.

2.) I would be interested for the authors to comment or speculate on why they see the substantive improvement in entangling gate versus the fidelity found in [28]. Could any of this be related to the choice of using cycle benchmarking versus interleaved randomized benchmarking?

3.) Is there a reference to Weyl twirling? I would like to see it included.

Reviewer #1 (Remarks to the Author):

This manuscript reports on high-fidelity qutrit entangling gates for superconducting transmon circuits. A microwave-activated cross-Kerr interaction is used to realize the qutrit entangling gates. The reported results constitute a good demonstration of logical operations on superconducting qutrits.

We thank the reviewer for spending the time to carefully read and provide comments on our manuscript.

My biggest reservation about the manuscript is that the main achievement is quantitative rather than qualitative. Some of the authors of this manuscript have previously reported qutrit entangling gates in references 28 and 32. In [28], the lowest infidelity was 11.1%. In this manuscript the lowest infidelity is 2.7%. This is an improvement. However, it is a quantitative improvement. It is true that the gate mechanism is different. However, this is not a major advance in the field.

On this point, we respectfully disagree with the reviewer. We would argue one of the main achievements presented in this work is the introduction and characterization of the differential-AC Stark shift for a tunable qutrit entanglement. Although the driven two-qubit-ZZ term generated by the differential AC Stark shift has been previously studied in a number of sources [refs. 39-42], the higher cross-Kerr terms generated by such an interaction have not. Employing this interaction to engineer an entangling gate with fidelity x4 better than the previous best attempt on a superconducting platform does constitute a quantitative improvement, but was only made possible due to the introduction of this new method of qutrit entanglement generation. Therefore, the gate mechanism in our work is not just different due to the new pulse scheme, but also due to the underlying physics. Prior attempts at this gate employed the parasitic, always-on dispersive coupling between two transmons to generate qutrit entangling phases; this former approach was both inflexible (as it cannot be tuned and is generally of the order ~ 100 KHz) and relied on an interaction (quantum crosstalk) that one would like to suppress in future qutrit systems. Our work allows for an entangling interaction in qutrits with a significant on/off ratio; we have added comments about this point to the introduction to help make this clearer. In our work, we are able to generate a maximally entangling gate on a 9 dimensional Hilbert space with a competitive fidelity to the state of the art 3 qubit gates, which operate on an 8 dimensional Hilbert space. The gates presented in this work employ a pulse scheme and tune up procedure that is not more complicated than many comparable qubit gates such as the Toffoli, which has many implementations that require using the qutrit levels anyways (such as e.g. refs. 34,37, and <https://arxiv.org/abs/2109.00558>). For these reasons, it is the feelings of the author that this work indeed constitutes a major advance for the practical implementation of qutrit processing with superconducting circuits..

Another concern is that qutrit logic is a niche research area. Most research in the field is focused on qubits because of obvious advantages such as simplicity and better coherence.

We agree with the reviewer that most of the research in the field is dedicated to studying qubits, as they are traditionally simpler to operate and easier to fabricate with high coherence. Nonetheless, we would like to point out our $\{|1\rangle, |2\rangle\}$ subspace coherence is already better or comparable to some qubits devices (i.e. the $\{|0\rangle, |1\rangle\}$ subspace) in the literature and available from industry (see e.g. Fig. 1 in ref. 31 or <https://qcs.rigetti.com/qpus>). Moreover, high-fidelity single qutrit operations have now been demonstrated in a number of works which we cite in the manuscript. So far, however, it is not clear that utilizing the third level and operating the system as a qutrit is advantageous compared to the simplicity of working with qubits. It is our feeling, though, that our work allows tuning up and operating qutrit entangling gates in a much simpler manner, opening the door to future efforts benefiting from the advantages of achieving control over a larger Hilbert space without added hardware or on chip overhead. It is our hope that this will help catalyze the already growing subfield of quantum information processing user qutrits.

Overall the manuscript is clearly written. However, there are some poorly written parts. Here are some examples. It is not explained what “more connected” means. The work is motivated by saying that realizing multi-qudit systems is challenging. However, if transmons are qudits, then any multi-transmon circuit is a multi-qudit system. So this motivation sounds weak. The manuscript says, without a proper justification, that the interaction in this work “generates entanglement between the entire two-qutrit Hilbert space”. The poor wording aside, there is no explanation for how this is different from previous work on two-qutrit gates. These problems can be fixed with a rewriting of the poorly written parts.

We thank the reviewer for these useful comments on the wording within the manuscript. We have updated the language within the manuscript to reflect our comments here. In the following, we respond point by point:

- *By “more connected” we mean that within a fixed quantum processor topology (i.e. a ring, a line, etc.) the amount of connected states for that processor is much higher in a qutrit processor rather than a qubit processor. For example, if we consider a line of 4 transmons with nearest neighbor coupling, then the graph of connected states with nearest neighbor coupling looks as follows:*

Similarly, using that same processor as qutrits, we will realize the following graph:

If we want to compare the same number of nodes in the same type of topology in both cases, then we can consider 3 qubits connected linearly to 2 qutrits (6 nodes in both cases) in which case we find 9 edges for the qutrits and 8 for the qubits. We also note that this point of a more connected structure for qutrits was also made in ref. 28.

- While it is true that transmons are naturally qudit objects, it has traditionally been difficult to realize control (especially when it comes to entanglement) over this qudit/qutrit Hilbert space. We would not consider any experiment using transmons as qubits to be a qudit experiment, because the higher states are not generally populated, readout, or coherently controlled in such a scenario. It is of course true that all multi transmon circuits are technically multi qudit circuits, but this is not useful if those states are not employed for computation; one of the main points we would like readers to take away from this work is that the higher $|2\rangle$ state can be a powerful resource for quantum processing.
- *Reproducing the entire sentence from the manuscript here*

“We note that this Hamiltonian generates entanglement between the entire two-qutrit Hilbert space, contrary to the CR case, where the entanglement is mostly restrained to a subspace of the qutrit [28, 48].”

What is meant here is that in the case of the cross-resonance effect for qutrits, the conditional Rabi oscillations are largely constrained to take place within a two level subspace of the qutrit, as can be noted from, e.g. Fig. 2 a and b in ref. [28]. In the case of the differential AC Stark shift, one can realize dynamic entangling phases simultaneously placed on the $|11\rangle$, $|12\rangle$, $|21\rangle$, and $|22\rangle$ states of the two qutrits. This is sufficient as a native entangling interaction to efficiently generate maximally-entangled qutrit C-phase gates as we performed in this work.

A technical point about the qutrit CZ gate is that it does not reduce to a qubit CZ gate if we look only at the states 0/1. So the basis for generalizing the CZ gate from qubits to qutrits warrants some justification. When equation 3 is introduced, it is not clear if it is taken from the literature or introduced for the first time in this manuscript.

We thank the reviewer for allowing us to make this discussion clearer in the text. The qutrit CZ comes directly from the generalization of the Pauli group employed in qubit based computation, to the Heisenberg-Weyl group employed for qudit/qutrit computation. Using this basis allows one to generalize the Clifford group as well as the Pauli group to qudits. This generalization of the qubit Pauli group to qutrits is discussed in Supplementary Note 3 of this work. The CZ gate follows directly from the definition of the Z gate in ternary based computation. For an additional reference, see equation 8 in ref 28. Notably, the CZ and CZ[†] gate performed here are locally equivalent to the CSUM (controlled sum) gate, which is the qutrit extension of the CNOT gate in qubits. We have updated the supplementary note 3 to additionally explain the unitarity of the CZ and CZ[†] gate and reference the supplementary note at this point of the main text.

Except for these issues, the manuscript reports good results in the area of superconducting qutrits.

We gratefully thank Reviewer #1 for their time and helpful comments on our manuscript. We hope that our response here and updates to the manuscript help alleviate any lingering concerns on the part of the reviewer about the manuscript.

Reviewer #2 (Remarks to the Author):

In the manuscript by Goss et. al., the authors demonstrated and benchmarked an all-microwave qutrit CZ gate using differential AC Stark shifts. This paper has many impressive aspects including a detailed experimental characterization of the cross-Kerr dynamics, and a clever use of echoes to construct the qutrit CZ gate without fine tuning the cross-Kerr dynamics. This work is an extension of differential AC Stark shifts in qubit systems [ref. 39-42]. The ability to generate

3-dimensional Bell states using a single qutrit CZ gate is attractive; potentially allowing one to generate 3-dimensional multipartite entangled GHZ states in transmons. This work deserves publication, but I'd like to see the following questions addressed first.

We gratefully thank Reviewer #2 for their time, helpful comments, and appreciation of our work contained within the manuscript. Here we respond point by point to the questions the reviewer would like to see addressed, and have updated the manuscript to reflect any relevant changes and clarifications discussed here.

1. Can the authors comment on how the charge noise in the 2 state impact the qutrit CZ gate? In the device data given in the supplementary, the T2 times for 12 and 02 levels are considerably lower than that of 01 levels.

In the case of, for example, the chip in Table S1 of the supplement, the ratio of EJ/EC is approximately ~ 73 . In this case, the charge dispersion of the $|2\rangle$ and $|1\rangle$ state is approximately ~ 10 KHz and ~ 250 Hz respectively (see the supplement of ref. [28]). We therefore expect that charge noise is largely suppressed in our system for the computational qutrit levels; nonetheless, we do see decreased dephasing of the higher levels, likely due to stronger matrix elements of the higher levels coupling to sources of dephasing (we also attribute this to the decreased T1_21 time). We additionally note that in the supplementary note 1, the T2_12 echos are much better than their Ramsey counterparts, so part of the advantage of our gate scheme is that it contains an effective echo pulse on the $\{|1\rangle, |2\rangle\}$ subspace of the qutrit.

2. I did not find a centralized description of the gate parameters for the qutrit CZ/CZ[†] gate. The gate times are in the caption of Fig 3, and the powers in the supplementary. What was the drive frequency and how were the frequency and power chosen?

We thank the reviewer for pointing out this omission, we have added the drive frequencies and approximate amplitudes for both gates to the tables in Supplementary note 1. Currently, to find workable parameters to perform both gates, the authors had to scan many different parameters of the Stark drive until finding the approximate conditions on the cross-Kerr parameters (rates of accumulation of qutrit entangling phases) discussed in the text; from there, fine tuning of the drive parameters was performed by implementing the pulse scheme for the gate and measuring the entangling phases.

3. Did the authors simulate terms beyond the 4 in the cross-Kerr Hamiltonian? In regions where the differential AC Stark fails the cross-Kerr Hamiltonian is presumably no longer a good approximation and other terms will be present.

Our master equation simulation included 4 levels for both of the transmons. In general, we tried to perform the interaction in regions where we expect the cross-Kerr

Hamiltonian to be a good approximation of the dynamics in our system by maintaining a large detuning from the resonances plotted in Figure 2. However, we do expect that transient two level systems and higher level transitions such as two photon transitions do impact our data. We discuss this further in Supplementary Note 8, but in general, we only included data that fit well to the linear accumulation of entangling phases one would expect under a cross-Kerr evolution.

4. Beyond the 4 resonances present in Fig. 2e, are there additional regions where the qutrit CZ gate does not perform well? For instance near the middle between $\omega_{01,a}$ and $\omega_{01,b}$ I would expect CZ dynamics to get complicated due to 2-photon transitions.

This is correct, two photon transitions would be a concern in this region. Thankfully, due to the flexible nature of this entangling interaction, we have the freedom to always place our Stark drive away from points of concern such as this. In other words, thanks to the flexibility of this entangling scheme, we do not have to perform the gate in such a region of concern.

5. In the paragraph below equation 4, the 2 approximated equations seem to imply $\alpha_{11} + \alpha_{22} \sim \alpha_{12} + \alpha_{21} \sim 0$. Can the authors double-check this?

We are sorry for the confusion, this was poorly worded in the original manuscript. Here, the two conditions on the cross-Kerr parameters correspond to two different conditions for two different gates, not two conditions that are simultaneously met. The first approximate equation ($\alpha_{11} + \alpha_{22} \sim \alpha_{12} + \alpha_{21}$) corresponds to the conditions met when experimentally performing the CZ unitary and the second to the CZ[†] unitary. We have made the main text more clear to avoid confusion.

6. In the same paragraph the authors state “the ramp feature can lead to offsets ... approximate relationship on the α_{ij} terms with some adjustment necessary”, what is the adjustment?

We thank the reviewer for this comment and the opportunity to add clarity to the wording in the main text. In our work, we employ a ramp up and down feature for the Stark drive in order to limit leakage and to ensure adiabaticity under the drive. This ramp leads to an offset on the linear accumulation of the entangling phases, as we only expect the linear α_{ij} rate of entangling phase to fit to the flat top part of the Stark drive. Due to this ramp feature, the gate time is not just set by the linear cross-Kerr parameters alone (which are found by fitting a linear model that includes an offset term to the accumulation of entangling phase for each α_{ij}). For tuning up the two-qutrit gate which has many conditions to be met, this means that we do not just want to find the exact conditions under equation 4 (with an equals rather than approximately equals sign). In practice, the “adjustment” has been looking for where those conditions are approximately met, then performing the actual gate scheme in Figure 3, and adjusting the pulse parameters until

the target entangling phases are most accurately met. We have expanded this discussion in the main text to add clarity to this point.

7. It might be helpful to have different labels for the phases ϕ_a ϕ_b and powers in Fig.2a and 3, to be consistent with Eq. 2 and Fig. 1.

Thank you for pointing out this inconsistency, we have corrected this in the text.

8. Why were the CZ and CZ[†] gates demonstrated on different samples?

This is mostly a story of availability on different measurement setups within our lab. The CZ gate was performed first on a measurement setup where we often cycle new chips in the ongoing development of quantum processors in our lab. That chip, for example, has already been replaced. The CZ[†] gate was performed later on a measurement set up and chip that is more stable within our lab, after much of the development of the experiment and physics had been worked out on the other experimental set up. We ended up performing the CZ[†] on this second set up partially because it was a different and complementary gate to the CZ, and partially because it corresponded to a condition on the cross-Kerr parameters we were easily able to satisfy with an early search of the cross Kerr parameters. We decided to include both gates in our work because we believe that together, they provide a fair accounting of the range of fidelity we can currently achieve with this gate schematic, and secondly because they demonstrate that multiple two-qutrit unitaries can be expressed with our underlying gate and entanglement scheme.

Reviewer #3 (Remarks to the Author):

Review of High-Fidelity Qutrit Entangling Gates for Superconducting Circuits

The authors implement High fidelity qutrit entangling gate that can provide the basis for qutrit-based quantum computers. This gate is implemented using a cross-Kerr resonant pulse. The gate designed appears to have higher entangling power than a Cinc or Cex gate. he work does establish significance for the field. It provides a proof of principle demonstration of a powerful entangling gate for qutrits. This demonstration of the gate likely can be used to help develop more efficient compilers for qubit architecture simulations such as those provided in Gokhale et al. "Asymptotic improvements for quantum circuits via qutrits." This work appears to be better than 82% fidelity result for a Cinc gate implemented by Morvan et al. in 2020.

I find the authors work novel and timely and warrants publication following the minor revisions.

We gratefully thank Reviewer #3 for their time, helpful comments, and appreciation of our work contained within the manuscript. Here we respond point by point to the questions the reviewer would like to see addressed, and have updated the manuscript to reflect any relevant changes and clarifications discussed here.

Requested Revisions:

I have a few revisions that I believe are necessary before approving for publication

1.) I would like to see a slightly wider discussion of the benefits qutrit architecture may apply to quantum algorithms. As the current connection to the benefits of qutrit hardware to algorithms seem sparse.

We thank the reviewer for the suggestion to expand this discussion in the text. We have updated the main text to make more explicit the benefits of qutrit hardware to algorithms.

2.) I would be interested for the authors to comment or speculate on why they see the substantive improvement in entangling gate versus the fidelity found in [28]. Could any of this be related to the choice of using cycle benchmarking versus interleaved randomized benchmarking?

The substantive improvement in the gate fidelity when compared to ref. [28] is due to the breakthrough of the differential AC-Stark shift that allows one to tune up the effective diagonal coupling (ZZ for qubits or cross-Kerr for qutrits as introduced in this work) between two transmons. This allowed us to significantly speed up the time of the two-qutrit gate by a factor of ~ 3 and to reduce the number of single qutrit pulses required to implement the proper unitary. This driving scheme was not known when ref. [28] was published. We also attribute this significant increase to the higher quality of coherence in our two devices. This was possible thanks to improvements in the fabrication of our device detailed in <https://iopscience.iop.org/article/10.1088/1361-6668/ab8617>. Finally, we want to point out to the reviewer that in ref. [32], the fidelity of the 2 qutrit gate was measured with Cycle Benchmarking as well. In general, Cycle Benchmarking provides a tighter bound on the fidelity of entangling gates than interleaved randomized benchmarking; additionally, to our knowledge, two qutrit randomized benchmarking has not been performed due to the large overhead in the required number of multi-qutrit gates per circuit.

3.) Is there a reference to Weyl twirling? I would like to see it included.

Thank you for pointing out that this reference was omitted in the main text. Weyl twirling was first experimentally performed in ref. 32 “Qutrit Randomized Benchmarking”, where it was referred to as Pauli twirling (as the Weyl group is just the qutrit generalization of the qubit Pauli group). We have updated the text to include this reference.

A tracked changes version of the manuscript is provided where all updates are marked with red text and red slashes through deleted phrasing. A summary of changes made to the manuscript is now summarized here:

1.) On page 1 in the introduction, originally it said:

“There are several proposals utilizing qutrits to improve quantum algorithms [17–20] and in the short term, qutrits can improve quantum simulation [21] and other NISQ applications [22].”

This was changed to:

Additionally, there are several proposals utilizing qutrits to improve quantum applications such as factoring with Shor’s algorithm [17], performing the quantum Fourier transformation [18], providing speedups for oracle based quantum algorithms [19], improving quantum simulation [20], and asymptotically improving algorithms such as Grover’s search [21, 22].

2.) On page 1 in the introduction, originally it said:

“Nonetheless, past qutrit entangling gates in transmons have been limited by relying on either a slow, static interaction or an interaction that restricts the entanglement to only a subspace of the qutrit.”

This was changed to:

“Nonetheless, past qutrit entangling gates in transmons have been limited by relying on either a slow, static interaction which can only be sped up at the expense of increased quantum crosstalk on the qutrit processor or an interaction that restricts the entanglement to only a subspace of the qutrit.”

3.) On page 1 and 2 in the introduction, we added:

“The tunable nature of this entangling interaction enables a large on/off ratio, allowing for future high-fidelity, simultaneous single-qutrit and two-qutrit operations in transmon qutrit processors.”

4.) On Page 2 we fixed a small error in Figure 1 where the capacitive coupling J had a slightly clipped edge from a figure that was overlaid improperly.

5.) On page 2, originally it said:

“Both gates performed in our work are universal for ternary computation and Clifford gates needed for QEC in qutrits.”

This was changed to:

“Both gates performed in our work are universal for ternary computation, maximally entangling, and Clifford gates needed for QEC in qutrits.”

6.) Figures 2 and 3 were updated so that the symbols in the Stark Drive portion of the pulse diagram were consistent with the notation in Eq.2 and Figure 1.

7.) On page 3 in the Results subsection titled “Qutrit CZ/CZ[†]”, we added:

“the CZ unitary follows directly from generalizing the qubit Pauli group to qutrits and is explained in further detail in Supplementary Note 3”

- 8.) On page 3 and 4 of the Results subsection titled “Qutrit CZ/CZ[†]”, originally it said:
“Tuning the experimental knobs demonstrated in Fig. 2, we are able to find the relaxed conditions on our crossKerr evolution: $(\alpha_{11} + \alpha_{22}) \approx -(\alpha_{12} + \alpha_{21})$ and $(\alpha_{11} + \alpha_{22}) \approx 2(\alpha_{12} + \alpha_{21})$. Under these conditions, at some drive time τ , we will have approximately acquired the desired entangling phases found in Eq. 3 to synthesize respectively a CZ or CZ[†] unitary. We find that when using flat-top cosine pulses for the cross-Kerr drive, the ramp features can lead to offsets on the linear accumulation of entangling phase; for this reason, the highest fidelity implementation of our gates only uses an approximate relationship on the α_{ij} terms with some adjustment necessary.”

This was changed to:

“Using the experimental knobs demonstrated in Fig. 2, we are able to satisfy approximate conditions on our cross-Kerr evolution that are ideal for compactly generating qutrit controlled phase gates. Specifically, we find drive parameters that satisfy $(\alpha_{11} + \alpha_{22}) \approx -(\alpha_{12} + \alpha_{21})$ for performing the CZ gate and $(\alpha_{11} + \alpha_{22}) \approx 2(\alpha_{12} + \alpha_{21})$ for performing the CZ[†] gate. The Stark drive parameters that meet these conditions are provided in Supplementary Note 1. Under these conditions on the cross-Kerr, and with the unitary evolution provided by our gate scheme in Eq. 4, at some drive time τ , we will have approximately acquired the desired entangling phases found in Eq. 3 to synthesize respectively a CZ or CZ[†] unitary. To ensure adiabaticity and limit leakage, we perform the Stark drive via flat-top cosine pulses with ramp up and down features. This ramp leads to effective offsets on the accumulation of the entangling phases, as we only expect the linear accumulation of entangling phase given by α_{ij} in Eq. 2 to correspond to the flat-top of the Stark drive. When tuning up the two-qutrit gates, this means that we first perform parameters sweeps to find regions where the previously mentioned conditions on the α_{ij} s are met, then perform the actual pulse scheme in Fig. 3 and adjust the Stark drive parameters until the target entangling phases given by Eq. 3 are most accurately acquired.”

- 9.) On page 8 we added a citation for Weyl-Twirling in the Results subsection “Benchmarking”.
- 10.) The Data Availability, Code Availability, Author Contributions, and Competing Interests statements were added to the text.
- 11.) The frequency and amplitude of the Stark Drives used to perform the CZ and CZ[†] were both added to supplementary note 1 of the text.
- 12.) An explanation for the form of the unitary for both the qutrit CZ and CZ[†] was added to supplementary note 5

REVIEWERS' COMMENTS

Reviewer #1 (Remarks to the Author):

I have read the authors' responses to my comments and the comments of the other reviewers. I appreciate the effort made by the authors to address all the reviewers' comments. I believe that the manuscript is well written and reports good research results. The authors' responses did not change my opinion of the significance of the achievement reported in the manuscript. However, the other two reviewers did not share my opinion. Taking their opinions into consideration, I can recommend acceptance of the manuscript at this time.

Reviewer #2 (Remarks to the Author):

I thank the authors for addressing the issues raised in the first report. I'm satisfied with the changes.

Reviewer #3 (Remarks to the Author):

I am content with the changes that the authors have made and recommend publication in nature.